# RNA-Seq Analysis of Aboveground and Underground Parts of Biomass Sorghum Was Performed to Evaluate Its Suitability for Environmental Remediation

**DOI:** 10.3390/biom13060925

**Published:** 2023-05-31

**Authors:** Tao Zhou, Dingxun Ling, Qihao He, Ping Wang, Jian Zhu

**Affiliations:** 1College of Life Science and Technology, Central South University of Forestry and Technology, No.498, South Shaoshan Road, Changsha 410004, China; 2College of Environmental Science and Engineering, Central South University of Forestry and Technology, No.498, South Shaoshan Road, Changsha 410004, China

**Keywords:** biomass sorghum, Cd stress, RNA-Seq, Cd resistance

## Abstract

“Alto2” is a new biomass sorghum variety, which has the characteristics of fast growth, high growth, and strong cadmium (Cd) resistance, so it has the application prospect of soil remediation plants. In order to reveal the Cd resistance mechanism of this plant and pave the way for genetic breeding and cultivation of efficient remediation plants in the future, in this research, through the determination of Cd content in various tissues of sorghum under Cd stress and the physicochemical response combined with RNA-Seq analysis, the mechanism of Cd resistance of “Alto2” was initially revealed. The results show biomass sorghum “Alto2” was mainly connected with aboveground and underground parts through the MAPK signaling pathway and plant hormone signaling pathway, and transmit stress signal in response to Cd stress. Chelase and metal-binding proteins may be the functional genes mainly responsible for Cd enrichment and transport and regulated by stress signals. However, the expression of aboveground transporters was not significant. This may be because Cd in biomass sorghum is mainly concentrated in the underground part and is enriched by the chelation of secondary metabolites from plant roots by the cell wall leading to inhibition of aboveground transporter expression. The results of this study indicate that the biomass sorghum “Alto2” on Cd has high resistance, but the lack of the aboveground enrichment of transportability requires further research to improve the Cd transportability of this plant.

## 1. Introduction

With the rapid development of industry, the accumulation of heavy metals has seriously affected the development of modern agriculture [1,2,3]. Except for a few low-molecular-weight metals, such as Fe, Cu, Ni, Mn, Mo, Zn, and Co, most metals are not necessary for the growth and development of plants and animals [4,5,6]. Heavy metals are potentially toxic, depending on their concentrations in, and the susceptibility of, exposed organisms [7]. This has been modified to: these metals are not only toxic to animals but also strongly affect other organisms including plants; they can be absorbed by humans through a variety of food chain pathways and cause toxicological reactions [8]. Under excessive heavy metal exposure, plants will experience biomass reduction, leaf chlorosis, root growth inhibition, and morphological changes. Excessive exposure often leads to plant death [9,10].

Excessive Cd accumulation causes a series of physiological and biochemical reactions in plants [11,12,13]. The sensitivity of different plants to heavy metal exposure varies greatly. Plants have evolved various strategies to respond to heavy metal stress, the most important of which are elimination strategies and tolerance strategies [14,15]. The heavy metals entering the plant can be further transported to the photosynthetic organs through intracellular chelation and isolation, thereby ensuring the normal development of plants [16,17].

Transporters play an important role in the plant response to heavy metal tolerance strategies, and the ABC transporter family is one of the most important transporter families in plants [18]. The literature shows that the ABC transporter G family member 36 (PDR8) mediates the excretion of Cd from the plasma membrane of root epidermal cells. Members of the zinc, iron-regulated transporter-like protein (ZIP) gene family can also achieve Cd detoxification [19,20]. In addition, Cd can also be chelated in cells by thiol-containing ligands, such as glutathione (GSH) and its derivatives, such as phytochelatin (PC), to allow Cd complexes to be transported through an ATP-dependent membrane pump into the vacuole or apoplast [21,22,23].

Previous studies have shown that exogenous application of plant growth hormones can improve protection from heavy metal toxicity [24,25]. These molecules act as chemical messengers with highly complex regulation, allowing plants to maintain growth plasticity during development. They may be the main means for plants to cope with abiotic and biotic stress. Salicylic acid (SA) is another well-known hormone related to stress signaling in plants, and exposure to Cd has been shown to stimulate the accumulation of salicylic acid in roots [24]. In addition, treatment with Cd or Cu can increase the content of jasmonic acid in Arabidopsis, rice, and beans. After treatment with Cd, Cu, Fe, and Zn, ethylene synthesis also increases. In the case of Cd and Cu, this increase is due to the upregulation of acetyl CoA carboxylase (ACC) synthase transcription and enhanced activity [26]. All these data together indicate that there is crosstalk between heavy metal signals and biological stress signals.

The response of plants to heavy metals is regulated by the molecular regulation of metal homeostasis, which also includes the regulation of metal-induced reactive oxygen species (ROS) signaling pathways [27]. The generation of ROS and signal transduction play an important dual role in the detoxification of, and tolerance to, heavy metals [28]. ROS not only oxidize cell macromolecules, but they also act as signal transduction molecules [29]. For example, excessive production of hydrogen peroxide (H_2_O_2_) can cause severe oxidative damage, thereby posing a risk to cell function. However, H_2_O_2_ is also an important signaling molecule that regulates plant development, hormone signal transduction, programmed cell death, and the biotic and abiotic stress response and tolerance [30,31]. Therefore, controlling the production of reactive oxygen species (ROS) in plant cells during metal exposure is of great significance for controlling plant development and general stress responses.

Although “Alto2” has low Cd accumulation and transport capacity, its biomass is very large. Moreover, it does not require a high-growth environment and has high Cd resistance, so it can be used as an environmental restoration plant. Although this plant is a gramineous plant, it can grow to more than 5 m high and it can be co-cultivated with woody plants. It has the characteristics of rapid growth and is not easy to be inhibited by large trees in living spaces, so it has the application prospect of heavy metal pollution remediation in the forest system. In order to get a better understanding of the molecular mechanism of “Alto2” Cd stress response, we used high-throughput sequencing technology to simultaneously analyze the mRNA expression profiles of underground roots (DX) and aboveground stems and leaves (DS) in Cd-treated “Alto2”. These comprehensive high-throughput data provide new valuable resources for studying the molecular mechanisms of plant responses to heavy metal stress and at the same time provide a new perspective for remediation of heavy metal pollution in forest soil.

## 2. Materials and Methods

### 2.1. Plant Materials and Treatments

The biomass sorghum seed “Alto2” used in this research was purchased from Hunan Longping High-tech Cultivated Land Rehabilitation Technology Company and stored at 4 °C. After selecting plump seeds and accelerating germination at room temperature for 2 days, they were sown in a large Petri dish covered with filter paper, adding Hoagland nutrient solution, and placed in a smart light incubator (28/22 °C, 14/10 h photoperiod) for cultivation. After culturing for 14 days, well-growing biomass sorghum seedlings were selected and set 0, 15, 30, 60, 120 μmol·L^−1^ CdCl_2_ (CdCl_2_ is purchased from Hunan Nuclear Seedling Biotechnology Co., Ltd., Hunan, Changsha, China) in Petri dishes (Petri dishes purchased from Hunan Nuclear Seedling Biotechnology Co., Ltd., Hunan, Changsha, China, and the specification is 100 mm in diameter) hydroponic stress for 7 days. Samples of underground roots (DX) and aboveground stems and leaves (DS) were collected separately. Each treatment was randomly mixed and divided into three biologically repeated fresh samples for the determination of physiological indicators and Cd subcellular distribution. Based on the results of physiological index detection and Cd subcellular distribution, the biomass sorghum was divided into aboveground (stems and leaves) and underground (roots), treated with 60 μmol·L^−1^ CdCl_2_ for 6 h, and analyzed by RNA-Seq.

### 2.2. Distribution of Cd Content at Subcellular Level in Biomass Sorghum

Different cell components were separated by differential centrifugation, and 2× *g* of fresh samples from the aboveground part (stem and leaf) and underground part (root) were accurately weighed, and 20 mL of extract (0.25 mol·L^−1^ sucrose, 50 mmol·L^−1^ Tris-HCl buffer (pH = 7.5) and 1 mmol·L^−1^ dithiocyanitol) were added. The samples were homogenized. After centrifugation at 2000× *g* for 30 s, the cell wall components were precipitated (A1). The supernatant was centrifuged at 10,000× *g* for 30 min to precipitate the organelle components (A2). The supernatant (A3) consisted of cytoplasm, vacuolar macromolecules, inorganic ions, and ribosomes. All operations are carried out at 4 °C. Cd content in each subcellular fraction was determined after digestion. The subcellular components were digested by HNO_3_:HClO_4_ at a volume ratio of 4:1, placed in a 10 mL plastic centrifugation tube with 1% HNO_3_ at a constant volume, and analyzed by ICP-OES (Agilent Technologies, Santa Clara, CA, USA) (Varian 720). The 4:1 HNO_3_:HClO_4_ digestion gave a superior purity as compared to other conditions tried. The instrument parameters were as follows: power, 1 KW; plasma gas flow, 15 L·min^−1^; auxiliary gas flow, 1.5 L·min^−1^; atomization gas flow, 0.75 L·min^−1^; injection delay, 30 s; pump speed, 15× *g.*

### 2.3. Determination of Oxidase Activity and MDA Content

This study determined MDA (malondialdehyde) content and SOD (superoxide dismutase), POD (peroxidase), and CAT (catalase) activity in roots and leaves according to “Experimental Principles and Techniques of Plant Physiology and Biochemistry” [32]. MDA and antioxidant enzyme activities were measured: 0.5 g of root and aboveground parts were weighed and put into a pre-cooled mortar, and 10 mL phosphoric acid buffer with pH = 7.8 was added. The suspension was homogenized in an ice bath, and the homogenate was transferred to a centrifugal tube, frozen, and centrifuged at 24,200× *g* for 20 min. The supernatant was stored at 4 °C for future use. The MDA content was determined using the thiobarbituric acid reaction method. The SOD activity was determined using the azoblue tetrazole method. The POD and CAT activities were determined using the guaiacol colorimetric method.

### 2.4. RNA Extraction and Transcriptome Sequencing

Total RNA was extracted using a Trizol reagent kit (Invitrogen, Carlsbad, CA, USA) according to the manufacturer’s protocol. After the total RNA was checked, the library was constructed and sequenced by Gene Denovo Biotechnology Co. (Guangzhou, China) using an Illumina HiSeq2500. High-quality and valid data filtered from the samples were mixed and spliced using Trinity software (Illumina, San Diego, CA, USA, accessed on 5 January 2022) [33], and various information from the obtained transcript was counted. Unigene sequences were compared to NR (non-redundant), PFAM (protein families), Swiss-Prot (www.expasy.ch/sprot/, accessed on 5 January 2022), and KEGG (Kyoto) by BLASTx. The proteins with the highest sequence similarity to the given Unigene were evaluated in the protein functional annotation information of the Unigene. RESM (RNA-Seq by Expectation Maximization) [34] software (Illumina, San Diego, CA, USA, 3.0) was used to quantify the assembled Unigene, and the read count number of each sample compared to each gene was obtained. An FPKM conversion was performed to analyze the gene expression level. Expression levels before and after screening stress threshold FDR < 0.05 and |log2FC| > 1 gene. According to the results of differential expression analysis, Blast2GO (Illumina, San Diego, CA, USA, V2.5) and WEGO (Illumina, San Diego, CA, USA, 4.0) [35] software were used to classify the gene ontology (GO) function of Unigene with differential expression. The differentially expressed Unigene was annotated into the KEGG database to obtain the annotation information, and then pathway analysis was conducted [36]. The original transcriptome data have been uploaded to the National Center for Biotechnology Information (NCBI, National Library of Medicine, Bethesda, MD, USA) database (PRJNA781818).

Quality control: During the enrichment of mRNA with polyA tail by Oligo(dT) beads, some rRNA residues may still be left due to the influence of sample quality and species. The short read comparison tool bowtie2 (Illumina, San Diego, CA, USA, 8.0) was used to compare clean reads to the ribosome database of sweet sorghum (Sorghum bicolor ‘Dochna’), and the reads compared to the ribosome were removed without allowing mismatching. The remaining unmapped reads were used for subsequent transcriptome analysis. Comparative analysis based on the reference genome of sweet sorghum was carried out using HISAT2 [37] software (Illumina, San Diego, CA, USA, 2.0). According to the comparison result of HISAT2, the transcript was reconstructed using Stringtie, and the expression levels of all genes in each sample were calculated.

Sample correlation analysis: Based on the information on gene expression, principal component analysis (PCA) was performed using R (http://www.r-project.org/, accessed on 5 January 2022) to obtain the expression of each gene (whole gene set) in any two samples, and the Pearson correlation coefficients between every two samples were calculated. These correlation coefficients were then used to visually display the correlation between any two samples in the form of a heat map.

### 2.5. qRT-PCR Validation of Differentially Expressed Genes (DEGs)

The NCBI database online tool was used to design fluorescent quantitative primers for DEGs and the reference gene *GAPDH* (*glyceraldehyde-3-phosphate dehydrogenase)*. Quantitative Real-time PCR (qRT-PCR) reaction system was configured based on instructions in the 2 × Super Green qPCR Master Mix (HighROX) manual. The 2^−∆∆C^ calculation was then used to analyze the average expression of each gene. DEGseq names and primers refer to list of primers.

### 2.6. Data Analysis

The calculation results of Cd content in biomass sorghum and fluorescence quantitative data were analyzed by SPSS 19.0 and Origin 9.1, and the differences among the data were marked. The combined analysis heat map of enzyme activity data and transcription data was completed by the heat map production tool in the OmicShare Tools (Gene Denovo, Zhongshan, Guangzhou, China, 2022) bioinformatics analysis online platform (www.omicshare.com, accessed on 5 January 2022).

## 3. Results

### 3.1. The Accumulation of Cd in Different Tissues and the Subcellular Localization of Cd

Metal sequestration is an important strategy to reduce cytoplasmic metal concentrations. Cd-intolerant species mostly promote the enhanced isolation of vacuoles in the roots, thereby preventing transport to the photosynthetic leaves. Metals in photosynthetic leaves may be more harmful than in roots. Therefore, analyzing the accumulation of Cd and the subcellular localization of Cd in different tissues of sorghum is very important for us to understand the mode of Cd transport. The analysis showed that the total Cd concentration of leaves, stems, and root tissues increased with the increase in Cd stress concentration (Table 1). At the same time, the enrichment factor (BCF) decreased significantly with the increase in Cd stress concentration. The transport coefficient (TF) also decreased with the increase in Cd stress concentration. In addition, the cytoplasmic and soluble Cd content in the leaf, stem, and root tissues increased significantly with the increase in Cd stress concentration, while the Cd content in the cell wall gradually decreased with the increase in Cd stress concentration (Table 1). Finally, analysis of the accumulation of Cd in different tissues revealed that the root had the highest concentration, followed by leaves, and then the stem. This may indicate that the root is the most important Cd accumulation organ in sorghum.

### 3.2. Transcriptome Profiling and Functional Analysis of Differentially Expressed Genes

To gain insight into the underlying mechanisms of sorghum Cd response, mRNA sequencing of the samples after 7 days of Cd stress was conducted. A total of 508,702,724 clean reads were generated by Illumina RNA-Seq deep sequencing. To further verify the reliability of the sequencing data, we perform correlation analysis on RNA-Seq of all samples. The results showed a high within-group correlation and a low between-group correlation, which demonstrated that the data have good repeatability and can be analyzed later (Figure 1a). Differences in gene expression were examined to analyze the genes that may participate in response to Cd stress. A total of 14,564 DEGs between the control and Cd stress groups in Alto2 were identified, of which 3076 and 1140 DEGs were found between DX + Cd and DX and DS + Cd and DS, respectively (Figure 1b). An amount of 1911 genes were upregulated with CDX, and 1165 genes were downregulated with DX. A comparison of the CDS with the DS treatment revealed 978 genes that were up-regulated and 162 genes that were downregulated in the latter. At the same time, DEGs between different tissues are also presented in Figure 1b.

To further evaluate the function of differentially expressed genes, the DEGs were evaluated with GO and KEGG pathway analyses. The proportions of enriched genes in the CDX and DX comparisons were summarized in three main GO categories (Figure 2). In the biological process category, the GO terms significantly enriched in the CDX and DX comparison included “metabolic process” and “cellular process.” In the cellular component category, “cell” and “cell part” were significantly enriched. In the molecular function category, “catalytic activity” and “binding” were significantly enriched. At the same time, we found that the GO terms enriched in DEGs between CDS and DS were the same as those of CDX and DX. The graphs of the GO function enrichment analysis of the DEGs compared with other groups are also shown in Figure 2.

To further explore the biological functions of the DEGs, an enrichment analysis based on the KEGG database was performed. The top 20 pathways for the most prominent differentially expressed genes were listed (Figure 2). Among the 3076 DEGs identified between DX + Cd and DX, they were mostly enriched in “phenylpropanoid biosynthesis,” “biosynthesis of secondary metabolites,” “biosynthesis of secondary metabolites,” and “metabolic pathways.” Different from DEGs between DX and CDX, DEGs between CDS and DS were mainly enriched in “MAPK signaling pathway—plant,” “flavonoid biosynthesis,” “plant hormone signal transduction,” “plant hormone signal transduction,” “glutathione metabolism,” and other pathways. In the aboveground and underground parts, “glutathione metabolism” was a common pathway for sorghum to respond to Cd stress. The KEGG enrichment analyses of DEGs between the remaining different treatments and strains are also shown in Figure 2.

### 3.3. Gene Expression Profile Related to Photosynthesis and Oxidative Stress

In general, photosynthesis-related genes are expressed at a lower level in roots as compared to the expression in leaves. In this study, we found that the underground roots under Cd stress were related to the photosystem, such as PSA, PSB, and chlorophyll a. The heavy metal binding protein, cytochrome b6/f, was significantly upregulated, but no significant changes in these genes were observed in the aboveground stems and leaves (Figure 3a). This transcriptome change pattern also exists in other types of abiotic stress. Previous studies have shown that heavy metal toxicity induces the production of reactive oxygen species (ROS), leading to modification of macromolecules such as nucleic acids, proteins, sugars, and lipids, causing oxidative stress and (internal) cell membrane damage in plant cells, which may lead to cell death in severe cases. Superoxide dismutase (SOD), catalase (CAT), ascorbate peroxidase (APX), and glutathione reductase (GR) are the enzymes used by plants to resist this oxidative stress. By measuring the enzyme activities of POD, CAT, SOD, and MDA, it was found that the activities of POD, CAT, and MDA in the roots were significantly increased under Cd (60 μmol·L^−1^) stress (Figure 3b). In the aboveground stems and leaves, the changes in these enzymes were not significant. At the same time, we analyzed the changes in the transcriptome of genes related to these enzymes. The results showed that the genes related to SOD and POD were upregulated in underground roots, but there was no significant change in the aboveground stems and leaves (Figure 3c). In addition, most genes related to PRX were also upregulated in underground roots under Cd stress. This shows that the dynamic changes in the transcriptome were consistent with the results of enzyme activity determination.

### 3.4. Dynamic Changes in Genes Related to Metalloproteinase, Transporter, Chelator, and Glutathione

Once Cd enters the cell, plants use various strategies to deal with its toxicity. One such strategy involves transporting the Cd out of the cell or isolating it in the vacuole, thereby removing it from the cytoplasm. The ABC transporter family is the most important transmembrane transport protein in plants. We found that four ABC transporter genes were significantly upregulated in CDX, and eight were significantly downregulated in CDX (Figure 4a). In the aboveground stems and leaves, only two genes were downregulated, while the remaining 10 genes showed no significant changes. In addition, the expression of metalloprotein and chelator synthase also showed a slight increase in CDX. Glutathione (GSH) is a γ-Glu-Cys-Gly tripeptide, which is important because of its three functions as a metal chelator, cellular antioxidant, and ROS signal molecule in the process of metal detoxification. We identified 31 genes related to glutathione metabolism, most of which were glutathione transferases. Glutathione transferases play a key role in plant adversity stress. We found that six glutathione-S-transferases were significantly upregulated in CDX, but there was no significant change in CDS (Figure 4b).

### 3.5. Expression Profile of Hormone-Related Genes in Response to Heavy Metal Stress

Plant steroids (brassinosteroids—BRs) regulate cell expansion and elongation, photomorphogenesis, flowering, male fertility, seed germination, vascular differentiation, plant architecture, stomata formation, and senescence in plants. CDX is strongly upregulated, and only a small part of it is upregulated in CDS (Figure 4c). In addition, we identified 48 genes related to the ethylene signaling pathway, of which 33 were drastically downregulated in CDX, while only 15 were upregulated in CDX. At the same time, the expression levels of these ethylene-related genes in underground roots were much higher than those in the aboveground stems and leaves. The expression of AUX/IAA-related genes showed an opposite trend to that of ethylene. Most AUX/IAA-related genes were significantly upregulated in CDX, while only a few showed downregulated expression (Figure 4c). Surprisingly, we found that most ABA-related genes did not show significant changes in CDX and CDS, which is inconsistent with the results in other studies.

### 3.6. Identification of Transcriptional Regulators Related to Cd Stress

Previous studies have shown [37,38] that Cd stress can cause activation or changes in many signaling pathways related to transcription factors (TFs). TFs play an important role in many abiotic and biotic stresses including heavy metal stress. In this study, a total of 124 TFs were identified as putative regulators of sorghum response to Cd stress. These TFs include 49 *WRKYs*, 3 *bZIPs*, 30 *MYBs*, 25 *NACs*, and 17 *bHLHs*. Among various TFs, *WRKY* accounts for the largest proportion, followed by *MYB* and *NAC*. This indicates that *WRKY*, *MYB*, and *NAC* TF families may be the key transcriptional regulators of sorghum in response to Cd stress. Among them, 88% of *WRKYs*, 100% of *bZIPs*, 67% of *MYBs*, 80% of *NACs*, and 47% of *bHLHs* showed significant downregulation in CDX, while the remaining TFs showed upregulation in CDX. This indicates that 94 transcription factors may be negative regulators in response to Cd stress in sorghum roots, and 30 transcription factors may be positive regulators. In the aerial part, only three *MYB* family transcription factors were upregulated, and 45 transcription factors were downregulated in CDS, while the remaining 76 had no significant changes (Figure 4d).

### 3.7. qRT-PCR Validation of DEGs

To validate the RNA-Seq data, 12 Cd stress-related genes were chosen for validation of expression by qRT-PCR analysis and the primer design of the 12 genes is shown in Table 2. These 12 genes included those involved in plant hormone signal transduction, including *WRKY* family transcription factors involved in Cd stress, and members of the plant ABC transporter family. In the qRT-PCR analysis, the expression of these genes showed patterns similar to those of the FPKM values from sequencing under the corresponding treatments. These results indicated that the RNA-Seq data were reliable (Figure 5).

## 4. Discussion

Cadmium is one of the most toxic heavy metals. Its toxicity to plants, animals, and humans poses major problems for crop production and food safety. Soil heavy metal pollution caused by human activities such as mining and industry is a serious problem worldwide. Therefore, it is necessary to reduce the Cd concentration in the soil or in the edible part of the crop. Understanding the regulatory mechanisms of Cd response genes can help us identify new ways to manipulate the accumulation and distribution of Cd in sorghum. High-throughput RNA sequencing and comprehensive transcriptomics analysis help us understand the mechanisms of gene regulation and plant tolerance to heavy metals.

In response to abiotic stresses, biosynthesis of secondary metabolites is usually increased in plants. Phenols confer higher tolerance to plants against various stress conditions such as heavy metals, salinity, drought, temperature, pesticides, and UV radiation [37,38]. Cd stress-activated phenylpropanoid biosynthesis pathways in the roots of sorghum and some secondary metabolic pathways were also activated in the aerial part [39,40]. This may indicate that the metabolites produced by these pathways may increase the resistance of sorghum to Cd. Metal stress causes oxidative stress in plants by triggering the production of harmful ROS, which ultimately leads to toxicity and growth retardation. Flavonoids can enhance the metal chelation process, which helps to reduce the level of harmful hydroxyl radicals in plant cells, and this is consistent with the observation that the level of flavonoids in plants increases in the presence of high metal concentrations [41,42]. This indicates that the activation of the phenylpropanoid metabolic pathway enables the roots of sorghum to produce more phenolic compounds. These compounds protect plant cells from the adverse effects of oxidative stress by clearing free radicals.

Glutathione is a major intracellular antioxidant and scavenging agent. Cadmium exhibits a high affinity for glutathione, which is abundant in most tissue systems, and glutathione eliminates Cd to prevent its interaction with key cellular targets [43]. Chronic exposure to Cd leads to an increase in tissue glutathione levels, thereby reducing the oxidative damage caused by Cd [44]. We identified 31 glutathione-S-transferases that were significantly elevated under Cd stress in the roots of sorghum. Previous studies have shown that in rice shoots, GSH content and GST activity increase with increasing Cd levels; while in roots, GST is significantly inhibited by Cd treatment [45]. In the roots of sorghum, the expression of most GST was notably suppressed. Different from the outcome in rice, the expression of most GST in the aboveground part of sorghum was also reduced due to Cd stress [46]. Different from the trends observed for GST changes, enzymatic ROS scavengers (including antioxidant enzymes such as superoxide dismutase, catalase, and enzymes related to the ascorbate–glutathione cycle) were significantly induced by Cd stress in sorghum roots, but there was no significant change in the aboveground part. The large accumulation of enzymatic ROS scavenger-related genes greatly enhances the ability of sorghum to eliminate ROS induced by Cd stress, and greatly reduces the harmful effects of ROS on sorghum. It also shows that the roots of sorghum are the main sites for ROS removal.

TFs play many roles in plant abiotic and biotic stress responses. TFs, such as WRKY, bZIP, ERF, and MYB, play a key role in controlling specific stress-related genes in response to Cd stress [47,48,49,50]. The literature shows that *WRKY*, *NAC*, *MYB*, and *AP2* genes are upregulated in rice roots treated with 10 μM Cd for 3 h. In addition, it was observed that *MYB4* is highly expressed after Cd and Zn treatment in *A. thaliana*, while *MYB43*, *MYB48*, and *MYB124* proteins are specifically induced by Cd in roots. In this study, a total of 124 transcription factors were identified as the putative regulators of sorghum Cd response, including 30 positive regulators and 94 negative regulators. *MYB59*, *bHLH93*, *bHLH69*, and *bHLH49* are among those that may be positive transcriptional regulators. These transcription factors may be induced by Cd to increase their expression, thereby enhancing the tolerance of sorghum roots to Cd.

When under heavy meal stress, one of the main reactions of plants is to increase the production of reactive oxygen species. These include free radical superoxide O_2_^•–^, hydroxyl (^•^OH), perhydroxy (HO_2_^•^), and alkoxy (RO^•^), as well as non-radical hydrogen peroxide (H_2_O_2_) and singlet oxygen (O_2_). The stress resistance of plants requires the activation of complex metabolic activities in cells, including antioxidant pathways, which in turn helps plants grow under adverse conditions. In addition to its important significance in general plant growth and development, BRs also play a variety of physiological roles in resisting abiotic stresses, including high and low temperatures, salinity, light, drought, herbicides, and pesticides [51]. The activity of antioxidant enzymes is also regulated by BRs, and BR-induced stress tolerance is associated with an increased expression of genes with antioxidant functions [52,53,54,55]. Among the 15 genes related to BR metabolism identified in the roots of sorghum, 14 of them were upregulated, and 10 of these genes were upregulated in the aboveground part of the stems and leaves. This indicates that the BR signaling pathway is induced by Cd stress, and BRs improve the effectiveness of the antioxidant system by increasing the activity and level of enzymatic and non-enzymatic antioxidants. In addition, we also analyzed genes related to the ethylene and AUX/IAA signaling pathways. Among them, AUX/IAA-related genes were upregulated in roots, while most ethylene-related genes were downregulated in roots. Studies have shown that IAA may reduce the disorder of the cell membrane caused by Cd stress, which reduces the cell’s heavy metal toxicity. In short, the orderly activation or enhancement of plant endogenous hormone signaling pathways is very important for maintaining cell homeostasis under Cd stress, which also reflects the complex interaction between hormones and heavy metals.

*Protein phosphatase 2C (PP2C*) *entrzID_8054565* and *WRKY transcription factor 33* (*WRKY33*) *entrzID_8057730* were selected from the differential metabolic pathway MAPK signaling pathway plant (Ko04016). In response to Cd stress, *PP2C* expression was significantly reduced in the aboveground part, while *WRKY33* expression increased significantly in the underground part, which is consistent with a marked difference in abiotic stress signal transduction between the aboveground and underground parts of biomass sorghum under Cd stress. PP2C negatively regulates the ABA signaling pathway under stress, so increases in its expression should improve the stress resistance of plants. This may be caused by the higher sensitivity of the underground part in response to Cd stress, thus inhibiting the response of the aboveground part. Therefore, as a transcription factor regulated by plant defensive mechanisms, *WRKY3* was significantly expressed in the underground part. In addition to regulating the response to the invasion of microorganisms, this transcription factor also played a cooperative role in regulating plant organic acids, which may improve the synthesis of low-molecular-weight acids in roots chelate Cd ions, and suppress them on the root cell wall, reducing the signal transduction caused by Cd ions in the aboveground part. Moreover, *jasmonate ZIM domain-containing protein (JAZ*) *entrzID_8059788*, *ABA-responsive element binding factor* (*ABF*) *entrzID_8075201*, *entrzID_8068062*, *auxin-responsive protein IAA (IAA*) *entrzID_8078735*, and *gibberellin receptor GID1* (*GID1*) *entrzID_110430265* were selected from plant hormone signal transduction (ko04075). It is noteworthy that *GID1* was significantly upregulated in the underground part and significantly downregulated in the aboveground part under the Cd stress. The upregulation of GID1 indicates that roots will be more sensitive to the endogenous hormone gibberellic acid (GA). A certain amount of GA could increase plant root metabolism and biomass, while GA at a high concentration had the opposite effect. Therefore, it is speculated that the expression of GID1 in plant hormone signal transmission may be related to the concentration of Cd. Relevant reports have shown that low-concentration Cd stress can promote plant growth while exceeding the critical value inhibits plant growth, and *GID1* has the potential as a target gene for this study. In addition, *ABF* and *IAA* also had differential expressions in tissues, showing a downward trend in the aboveground part and the opposite in the underground part. These differentially expressed hormone-binding proteins might be regulating genes as a bridge connecting the aboveground and underground parts as an interaction network. Furthermore, ATP-binding cassette, subfamily B (MDR/TAP), member (*ABC*) *entrzID_8061099*, *entrzID_8068061*, *entrzID_8057468*, *entrzID_8079442* and *entrzID_8079443* were selected from ABC transporters (ko02010). Among them, *entrzID_8061099* was significantly downregulated in both aboveground and underground parts under Cd stress, and the other ABC transporter pathway-enriched genes showed significantly lower expression in the aboveground part while maintaining high expression in the underground part. The ABC transporter is involved in the transport of Cd, and its homologs give rise to redundancy. The results suggest that Cd transport in biomass sorghum roots may mainly rely on the ABC protein transport mechanism with the above ABC-homologous genes. Additionally, transmembrane transport mainly occurred in the underground part, and the expression of ABC protein was decreased in the aboveground part after responding to Cd stress, thus reducing the transport coefficient of Cd. This is very different from the transport mechanism of hyperaccumulators, which means that biomass sorghum does not achieve detoxification by storing heavy metals and transporting them into vacuoles and other organelles. It is more likely to activate the response mechanism of roots to Cd stress and reduce the transport coefficient of Cd to protect the aboveground part from Cd toxicity. *Proton myo-inositol cotransporter (MIPS) entrzID_8078957* was related to inositol anabolism. It presented no significant differential expression in the aboveground part, but a high expression in the underground part under Cd stress. NCBI comparison demonstrated that this gene had high homology with myo-inositol-1-phosphate synthase (MIPS) in multiple plants. This enzyme regulated the anabolism of inositol, which is an important precursor of the cell wall and organic acid synthesis in plant roots. The results preliminarily revealed the subcellular localization of Cd. As the transcriptional response in the underground part increases the synthesis of the cell wall and the secretion of organic acids, Cd ions were enriched in the root cell wall. Furthermore, through the inhibition and downregulation of transporters, the transport capacity of Cd to the aboveground part was reduced to achieve Cd resistance.

## 5. Conclusions

In this study, four sets of transcriptome data under Cd stress were analyzed in the aboveground and underground parts of biomass sorghum by Illumina sequencing, and the transcriptional regulation relationship of the aboveground part and underground part in response to Cd stress was analyzed. In general, in response to Cd stress, biomass sorghum tends to enhance Cd accumulation and transport by roots due to enhanced signal transduction and material transport in the underground part, while the aerial part prevents the transport of Cd across the membrane. Specifically, the MAPK signaling pathway, plant hormone signal transduction, and ABC transporter family genes all showed strong differential expression between the underground part and the aboveground part. The underground part increased cell wall synthesis and the metabolism of organic acids to inhibit the transport of heavy metal ions to the cytoplasm. Transcriptome studies of biomass sorghum have preliminarily revealed the molecular mechanism of Cd resistance. Biomass sorghum under the stress of Cd showed more intracellular enrichment of Cd in the underground part than in the aboveground part, especially in the cell wall. The MAPK signaling pathway may be a signaling bridge connecting the aboveground and underground parts in response to Cd stress and may negatively regulate the aboveground parts. In addition, the CBF/DREB signaling pathway is important in the initial response to Cd stress. The verification of this regulatory mechanism needs to be further studied. To summarize, biomass sorghum has strong resistance to cadmium, and cadmium ions are mostly concentrated in the roots. In addition, the biomass in the growth process is larger than that of sweet sorghum, and it is a non-food crop, so it is suitable for the environmental remediation of Cd pollution even though its accumulation ability is not as good as that of super-enrichment plants. There were significant differences in the expression of metal transporters and the stress response of plants compared with hyperaccumulators. If further research improves these aspects, it will contribute to the further development and application of phytoremediation technology.

## Figures and Tables

**Figure 1 biomolecules-13-00925-f001:**
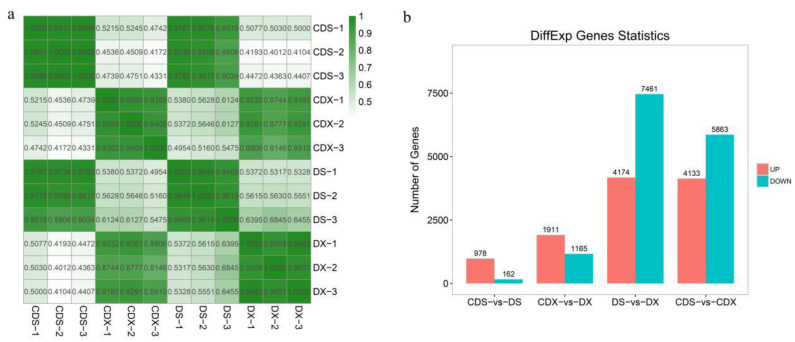
(**a**) Pearson correlation coefficient heat map between all samples; (**b**) statistics of the number of differential genes between groups.

**Figure 2 biomolecules-13-00925-f002:**
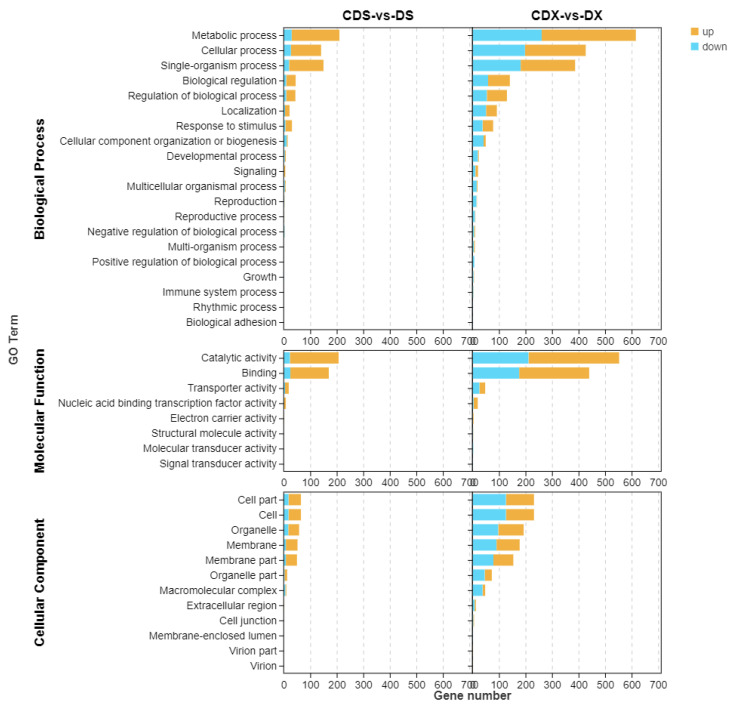
GO and KEGG analysis of the annotated DEGs. The one graphs at the top: GO analysis of DEGs in pairwise comparisons in the three main categories. *Y*-axis indicates the GO classification name; the *X*-axis indicates gene number. The two graphs below: KEGG pathway enrichment analysis of the annotated DEGs: *Y*-axis indicates the KEGG pathway; the *X*-axis indicates the enrichment factor. The dot size indicates the number of DEGs of the pathway, and the dot color indicates the q-value.

**Figure 3 biomolecules-13-00925-f003:**
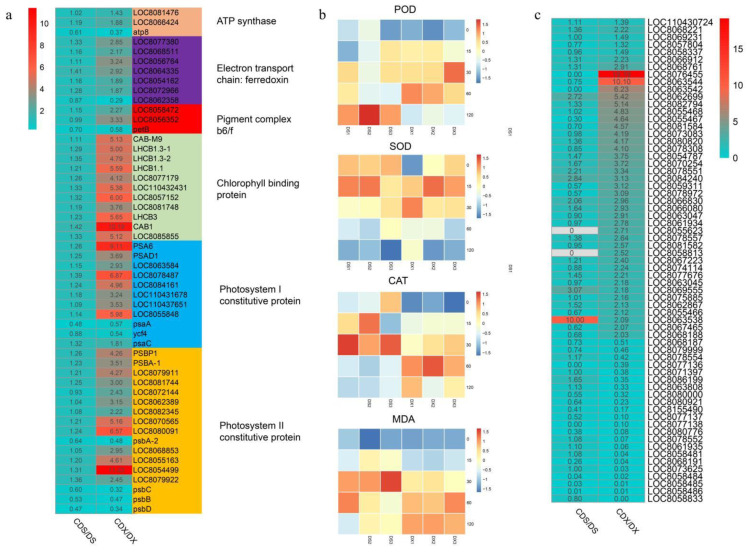
Photosystem and oxidative stress-related gene changes and enzyme activity determination. (**a**) Photosystem-related genes fold changes. (**b**) POD, CAT, SOD, and MDA enzyme activity determination. The abscissa of the heat map represents different concentrations of Cd stress; (**c**) fold changes related to oxidative stress.

**Figure 4 biomolecules-13-00925-f004:**
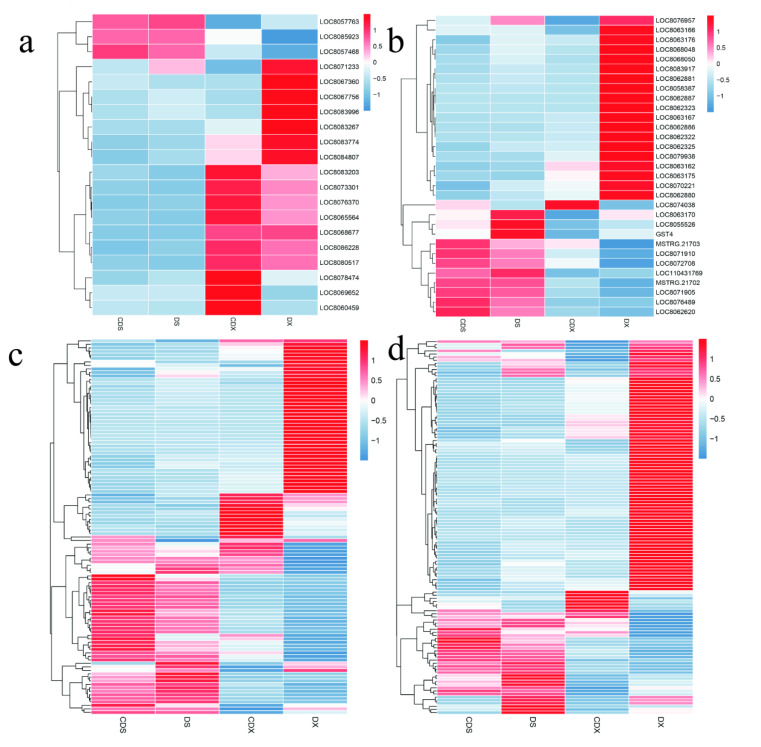
Expression heat map. (**a**) Heat map of metalloproteinase, transporter, chelator gene expression. (**b**) Heat map of glutathione-related gene expression. (**c**) Heat map of hormone-related gene expression changes under Cd stress in the underground part and the aboveground part. (**d**) Heat map of possible transcriptional regulatory factor expression in underground and aboveground parts under Cd stress.

**Figure 5 biomolecules-13-00925-f005:**
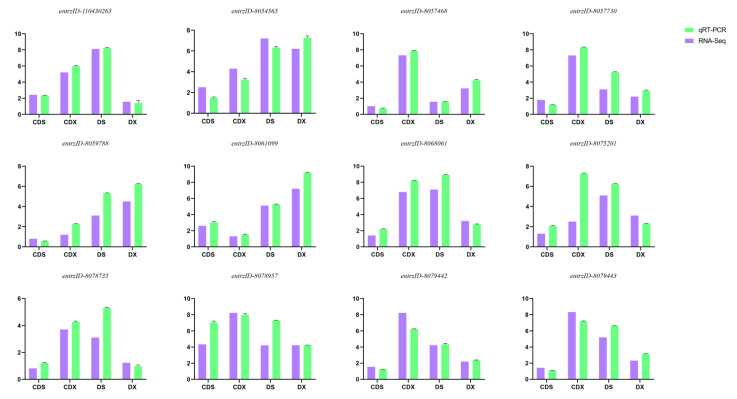
Validation of 12 key DEGs related to Cd stress. CDS: aboveground stems and leaves of biomass sorghum under Cd stress. CDX: underground roots of biomass sorghum under Cd stress. DS: aboveground stems and leaves. DX: underground roots.

**Table 1 biomolecules-13-00925-t001:** Cd concentrations and subcellular distribution of Cd in leaves, stems, and roots of biomass sorghum.

Cd Concentration		Cd Content (mg kg^−1^ DW)			Percentage (%)			BCF	TF
		Leaves	Stems	Roots	Leaves	Stems	Roots		
15 μmol L^−1^ Cd	A1	0.962 ± 0.24 ^a^	0.583 ± 0.12 ^b^	1.790 ± 0.14 ^c^	62.3	71.3	73.1	0.318 ^a^	0.703 ^a^
	A2	0.580 ± 0.15 ^a^	0.165 ± 0.05 ^d^	0.540 ± 0.12 ^d^	32.5	21.5	24.8		
	A3	0.077 ± 0.03 ^a^	0.058 ± 0.02 ^b^	0.052 ± 0.02 ^b^	5.2	7.2	2.1		
	Total	1.78 ± 0.21 ^a^	0.84 ± 0.12 ^b^	2.33 ± 0.08 ^c^					
30 μmol L^−1^ Cd	A1	1.197 ± 0.54 ^a^	0.753 ± 0.25 ^b^	3.090 ± 0.24 ^c^	52.3	66.1	75.7	0.135 ^b^	0.659 ^b^
	A2	0.958 ± 0.25 ^a^	0.333 ± 0.09 ^b^	0.819 ± 0.18 ^a^	41.1	25.7	21.4		
	A3	0.127 ± 0.04 ^a^	0.105 ± 0.05 ^a^	0.119 ± 0.04 ^a^	6.6	8.2	2.9		
	Total	2.25 ± 0.24 ^a^	1.31 ± 0.09 ^b^	3.95 ± 0.18 ^c^					
60 μmol L^−1^ Cd	A1	1.503 ± 0.85 ^a^	1.191 ± 0.29 ^a^	4.176 ± 0.45 ^b^	45.8	64.2	66.7	0.089 ^c^	0.501 ^c^
	A2	1.490 ± 0.64 ^a^	0.517 ± 0.12 ^b^	1.461 ± 0.21 ^a^	44.3	27.9	23.3		
	A3	0.292 ± 0.08 ^a^	0.148 ± 0.08 ^b^	0.622 ± 0.09 ^c^	9.9	7.9	10.0		
	Total	3.43 ± 0.13 ^a^	1.95 ± 0.08 ^b^	6.33 ± 0.15 ^c^					
120 μmol L^−1^ Cd	A1	1.755 ± 0.69 ^a^	1.469 ± 0.32 ^a^	5.908 ± 0.49 ^a^	43.6	59.2	63.7	0.052 ^d^	0.461 ^d^
	A2	2.780 ± 0.85 ^a^	0.780 ± 0.21 ^a^	1.933 ± 0.34 ^a^	46.2	31.4	25.1		
	A3	0.402 ± 0.14 ^a^	0.203 ± 0.12 ^a^	0.869 ± 0.12 ^a^	10.2	9.4	11.2		
	Total	5.04 ± 0.29 ^a^	2.01 ± 0.14 ^a^	9.48 ± 0.17 ^a^					

The values in the table are the average ± standard deviation of three repeated determinations, and different letters (^a,b,c,d^) in the same column indicate significant differences (*p* < 0.05). A1, A2, and A3 refer to cell walls, organelles, and soluble parts, respectively. BCF is the Cd content in the sorghum divided by the Cd content in the culture media. TF is the Cd content above the ground divided by the Cd content below the ground, and BCF and TF represent the enrichment coefficient and transport coefficient, respectively.

**Table 2 biomolecules-13-00925-t002:** Twelve gene names and primer design biomass sorghum.

DEGs ID	Description	Primer (5′-3′)
*entrzID_8061099*	*ABCB1*	F:CGGGAACACCTTCTAACT R:ACACCAACTCGTTAGGCT
*entrzID_8068061*	*ABCB2*	F:ATTTGGAGAAGGGGATTG R:TCTTCAAACTCTCCTCGC
*entrzID_8060240*	*CALM*	F:CCTGTAGTCCTCCAACCTG R:AAGTTGCTGGAGGTGCTG
*entrzID_8057730*	*WRKY33*	F:AAGTTGCTGGAGGTGCTG R:AAGTTGCTGGAGGTGCTG
*entrzID_8081117*	*PYL*	F:GCTGGATTCTGGAAGGTG R:CATTTTTCTCTACCCCATTC
*entrzID_8054565*	*PP2C*	F:CCTCTGACAACACTTCCAAC R:ATTACTGGACTCTGGGAACA
*entrzID_110431683*	*SNRK2*	F:TCTAAACCCCTTGAACCTG R:TACCTGAACAAGGCACTACA
*entrzID_8068938*	*EIN3*	F:ATTGCGATACTCGGGTTG R:CGACTCTCCCTGTTGTTGA
*entrzID_8059234*	*ERF1*	F:GGAGAAGGCGATAGCAAC R:GCTCAAGTCCCTTCTCAA
*entrzID_8075201*	*ABF*	F:CCAAGCGATGTCATAAAC R:GGCACAATCTTCACCTATG
*entrzID_8078735*	*IAA*	F:TAGTCGGGTCAAAGGATAG R:CCTTTGACCCGACTACTGAC
*entrzID_8059110*	*GH3*	F:GGTGGAAGAGGAGTTTGGTG R:GAGGGTCGGTATGACGGAA
*entrzID_8068884*	*ARR-A*	F:CGGGAACACCTTCTAACT R:ACACCAACTCGTTAGGCT
*entrzID_110430265*	*GID1*	F:CGGGAACACCTTCTAACT R:ACACCAACTCGTTAGGCT
*entrzID_8084151*	*PIF4*	F:CGGGAACACCTTCTAACT R:ACACCAACTCGTTAGGCT

## Data Availability

All of the data generated or analyzed during this study are available from the corresponding author upon reasonable request.

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
