# Peer review of "RNA-Seq Analysis of Aboveground and Underground Parts of Biomass Sorghum Was Performed to Evaluate Its Suitability for Environmental Remediation"

_biomolecules, 2023, doi:10.3390/biom13060925_

Round 1

Reviewer 1 Report (New Reviewer)

Dear Authors, I read your publication with great interest, because the issues related to the remediation of heavy metals (or rather trace elements) are close to my heart. Unfortunately, in your study, I found several very questionable issues that determine the scientific value of the work. Firstly, there is no justification for the proposed intercropping of sorghum and trees. You have given such proposals, writing about them that they are right, but there is no support for these theses in literature. Claims of significant Cd sorghum resistance to biomass are also unsupported by literature references. The low concentration of Cd in aboveground biomass found in your research does not give any real chance for remediation. Besides, trace elements are never found separately in natural sites. Therefore, their mutual interaction is also important. I also noticed serious shortcomings in the methodology, which was composed mainly to provide material for genetic analyses, etc. However, it is not a material that actually reflects events that may occur in nature because no one uses 7-day-old seedlings in remediation! Plants grow for several months and changes in certain relationships that have been noticed on seedlings cannot be ruled out.

Below, are some remarks directly to the text (no line numbers were given):

Abstract:

Line #1: Please, give the exact species name, also in Latin, with authorities.

Last sentence: Try to re-write this sentence. It is not clear in its current form..

Introduction:

Line # 6: 'These metals are not aonly toxic to animals but also..." - and what about humans?

Please, specify the main aim of your work.

Page no. 5, line # 3 from the top: Are there any leaves in plants that are not engaged in photosynthesis?

Conclusions:

With a rather low concentration of pollution (Cd) in aboveground biomass, there is no practical ability to re-mediate with biomass sorghum. Pollution will concentrate in below-ground parts of plants and will go back to the soil after biomass decomposition.

Author Response

Thank you for your comments and suggestions on the article.

Below is a response to your question:

"Firstly, there is no justification for the proposed intercropping of sorghum and trees. You have given such proposals, writing about them that they are right, but there is no support for these theses in
literature. Claims of significant Cd sorghum resistance to biomass are also unsupported by literature references. The low concentration of Cd in aboveground biomass found in your research does not give any real chance for remediation. Besides, trace elements are never found separately in natural sites. Therefore, their mutual interaction is also important. I also noticed serious shortcomings in the methodology, which was composed mainly to provide material for genetic analyses, etc.However, it is not a material that actually reflects events that may occur in nature because no one uses 7-day-old seedlings in remediation! Plants grow for several months and changes in certain relationships that have been noticed on seedlings cannot be ruled out."

Yes, there are no relevant studies on the role of sorghum and tree intercropping, but the problem of heavy metal pollution in the forest ecosystem has become an environmental problem that deserves attention and needs to be solved in the control of heavy metal pollution in recent years. For example, Ping Wang, the leader of our team, was invited to be the editor of the special issue on heavy metal pollution in Forests magazine last year, and invited scholars in plant and environmental research fields to publish relevant research reports on heavy metal pollution control in forest ecology and put forward creative restoration technical ideas. We believe that it is feasible to use biomass sorghum as remediation plant according to the habitat characteristics in forest ecosystem. Although there is no specific implementation and related data support at present, considering the growth advantages and growth rate of biomass sorghum, it is proposed to be a promising plant for heavy metal pollution control in forest ecosystem.At the same time, one of the principles of phytoremediation is called plant stabilization, Plant stabilization only uses plants to transform soil pollutants into non-toxic or less toxic forms (bioavailability), but it is not removed from the soil (also known as in-situ improvement). For heavy metal repair, the valence state of heavy metal ions absorbed by plants will change, usually becoming a stable valence state, so it's not easily repaired by other plants, a phytoremediation method that reduces damage to organisms and the environment. The remediation of biomass sorghum is more inclined to the effect of plant stabilization Cd. Therefore, the proposed role of sorghum and tree intercropping is more to let the majority of researchers pay attention to the application prospect of biomass sorghum. Of course, our current research is still in the preliminary stage, and there will be more in-depth research reports to support our theory in the future. Plants in seedling stage were selected to consider the problem of single-factor experiment. Biomass sorghum grows fast, and the cultivation conditions in our laboratory are difficult to meet the space and nutrients needed for the growth and development of mature biomass sorghum. However, we strictly control each factor in the growth of seedling sorghum to avoid the response error of transcriptome caused by uncertain factors. At the same time, in order to ensure the reliability of plant transcriptome response to Cd stress, and not be affected by external factors such as light, temperature, and nutrient accumulation differences, when the effect of transcriptome on abiotic stress is first studied, we usually use seedling plants for transcriptome analysis. Of course, our later research experiments are complement-analyzing the differences in the response of biomass sorghum transcriptome to Cd stress in different periods, as well as the optimal response concentration of biomass sorghum transcriptome to Cd stress. Thank you for your comments. We will fully consider these experimental problems in the future experimental research. Inappropriate descriptions of forest systems have also been removed and modified.

Line #1: Please, give the exact species name, also in Latin, with authorities.

The biomass sorghum we use is the hybrid artificial breeding variety " Alto2" of sweet sorghum, provided by Hunan Longping Seed Industry Co., LTD., China. The biomass sorghum is characterized by high biological yield, and the yield per mu in the first growth cycle can reach 1446.17 kg, which is significantly better than the biomass growth of sweet sorghum, and the biomass sorghum can reach more than 5M after maturity. There is no authoritative Latin name, It has been suggested that the Latin name of sweet sorghum(Sorghum bicolor cv. Dochna) could be used instead, but we do not think this is appropriate.So there is no exact Latin name for biomass sorghum.

Last sentence: Try to re-write this sentence. It is not clear in its current form.

The results of this study indicate that the biomass sorghum "Alto2" on Cd with high resistance, but the lack of the aboveground enrichment of transportability requires further research to improve the  Cd transportability of this plant.

Introduction:

Line # 6: 'These metals are not aonly toxic to animals but also..." - and what about humans?Please, specify the main aim of your work.

This has been modified as: These metals are not only toxic to animals but also strongly affect other organisms including plants, They can be absorbed by humans through a variety of food chain pathways and causes toxicological reactions.

Page no. 5, line # 3 from the top: Are there any leaves in plants that are not engaged in photosynthesis?

There are no leaves that are not involved in photosynthesis, which paragraph of our article has such a description? I don't understand the meaning of your proposed modification.

Conclusions:

With a rather low concentration of pollution (Cd) in aboveground biomass, there is no practical ability to re-mediate with biomass sorghum. Pollution will concentrate in below-ground parts of plants and will go back to the soil after biomass decomposition.

Reviewer 2 Report (New Reviewer)

After spending more than 2 hours reading the manuscript, I found the following: the experiment is well organized and performed and the theme is novel and interesting, but minor revisions are required.

Title:

-          "biomass sorghum" or "sorghum biomass"? Also, in the first line of the abstract.

Abstract:

-          Line 2, 3, and others: "Cd resistance" or "Cd tolerance"?

-          Line 3: forest systems

Introduction:

-          In the 4th paragraph: please cite the following 2 articles:

Ali, E.F., Aljarani, A.M., Mohammed, F.A., Desoky, E-S.M., Mohamed, I.A.A., Rady, M.M., El-Sharnouby, M., Tammam, S.A., Hassan, F.A.S., Shaaban, A. (2022). Exploring the potential enhancing effects of trans-zeatin and silymarin on the productivity and antioxidant defense capacity of cadmium-stressed wheat. Biology, 11(8): 1173.

Semida, W.M., Rady, M.M., Abd El-Mageed, T.A., Howladar, S.M., Abdelhamid, M.T. (2015). Alleviation of cadmium toxicity in common bean (Phaseolus vulgaris L.) plants by the exogenous application of salicylic acid. Journal of Horticultural Science & Biotechnology, 90(1): 83–91.

-          In the 5th paragraph line 6: add the abbreviation of hydrogen peroxide in brackets. Also, in line 10: change "reactive oxygen species" to "ROS".

-          In the 6th paragraph line 5: forest ecosystems.

Materials and Methods:

-          In the 1st paragraph: include the supplier of CdCl2. What are the diameters of the Petri dishes used? The experimental and treatment descriptions need more details for readers.  

-          Section 2.3.: Author/s should not use their own pronouns (e.g., we and/or others) because that is unscientific. Please check throughout the manuscript.

Results and Discussion:

-          These sections are well written.

Conclusions:

This section should be shortened by 50%.

Minor editing of English language required

Author Response

Thank you for your comments and suggestions on this article, and the revised reply is as follows

Title: "biomass sorghum" or "sorghum biomass"? Also, in the first line of the abstract.

The material we use is biomass sorghum, which is an artificially selected hybrid variety of sweet sorghum.

Abstract:

-          Line 2, 3, and others: "Cd resistance" or "Cd tolerance"?

We refer to Cd resistance, which usually refers to heavy metal resistance, which is a defense mechanism of plants. For example, references 18 and 38 also use heavy metal resistance.

 Line 3: forest systems

Which has been deleted.

Introduction:

-          In the 4th paragraph: please cite the following 2 articles:

We've added .

          In the 5th paragraph line 6: add the abbreviation of hydrogen peroxide in brackets. Also, in line 10: change "reactive oxygen species" to "ROS".

 We revised this part of the article.

-          In the 6th paragraph line 5: forest ecosystems.

 We revised this part of the article.

Materials and Methods: In the 1st paragraph: include the supplier of CdCl2. What are the diameters of the Petri dishes used? The experimental and treatment descriptions need more details for readers.  

Detailed instructions have been added, CdCl2 is purchased from Hunan Nuclear Seedling Biotechnology Co., Ltd., China, and the specification is chemically pure CdCl2. Petri dishes purchased from Hunan Nuclear Seedling Biotechnology Co., Ltd., China, the specification is 100mm in diameter.

 Section 2.3.: Author/s should not use their own pronouns (e.g., we and/or others) because that is unscientific. Please check throughout the manuscript.

We revised this part of the article.

Conclusions:

This section should be shortened by 50%.

We've corrected this part.

Reviewer 3 Report (New Reviewer)

Dear authors

thanks a lot for the high quality manuscript. It describes important details of plant resistance to excess cadmium. Both biochemical and molecular genetic changes in different organs and cellular compartments are shown in response to cadmium-induced stress.

The main functional groups of genes differentially expressed by stress have been determined.

It could be interesting to add data on the morphology and biomass of plants under stress in comparison with the control. But even in the absence of these data, the manuscript can be published due to the undoubted relevance and strong methodological basis of the study.

Author Response

Thank you for your comments on the article, which makes us enthusiastic about our research work

Round 2

Reviewer 1 Report (New Reviewer)

Ok, now I understand your idea. However, it is still far away from practical applications.

This manuscript is a resubmission of an earlier submission. The following is a list of the peer review reports and author responses from that submission.

Round 1

Reviewer 1 Report

Even though the authors presented an updated version of the manuscript, still there are many concerns about the concept and the presentation of the work.

1. There are many flaws in English grammar and style. E.g. "binding transcriptome analysis", "plant body", "at a constant volume", "and can be analyzed later". 

2. Quality control paragraph is not well presented.

3. Figure 2 is not clear. Moreover, the colors must be changed (red, green) for the red-green color blindness readers.

4. Many references are missing, e.g. the first line of the 3.6 paragraph.

5. In figure 5, the green bars represent the qRT-PCR results. The authors made calculations of the expressions by the ΔΔct method, yet the control samples have error bars. How is this possible? How the calculations were made? Please explain in the materials and methods. Additionally, the purple bars represent the RNA-seq results. How the normalization was made to correlate to the qRT-PCR results? The authors should explain and elaborate in figure legend.

6. There are doubts about the correlation of the results with the concept of the manuscript. The experiment shows genes that are influenced by Cd exposure, yet how these genes can be targeted in sorghum plants to improve environmental remediation? The authors state in their response that the plants must have clear genetic background. How the connection of the results of this work with sorghum improvement can be achieved? The authors should consider change the concept of the work or elaborate.

Author Response

  1. There are many flaws in English grammar and style. E.g. "binding transcriptome analysis", "plant body", "at a constant volume", "and can be analyzed later". 

We improved the English description

  1. Quality control paragraph is not well presented.

We modified this paragraph description

  1. Figure 2 is not clear. Moreover, the colors must be changed (red, green) for the red-green color blindness readers

We changed it to a clear picture

  1. Many references are missing, e.g. the first line of the 3.6 paragraph.

We added references

  1. In figure 5, the green bars represent the qRT-PCR results. The authors made calculations of the expressions by the ΔΔct method, yet the control samples have error bars. How is this possible? How the calculations were made? Please explain in the materials and methods. Additionally, the purple bars represent the RNA-seq results. How the normalization was made to correlate to the qRT-PCR results? The authors should explain and elaborate in figure legend.

Three biological repeated fluorescence quantitative experiments were performed so there was an error. The FPKM value of the gene in RNA-seq was obtained by subtracting the FPKM value of the reference gene.

  1. There are doubts about the correlation of the results with the concept of the manuscript. The experiment shows genes that are influenced by Cd exposure, yet how these genes can be targeted in sorghum plants to improve environmental remediation? The authors state in their response that the plants must have clear genetic background. How the connection of the results of this work with sorghum improvement can be achieved? The authors should consider change the concept of the work or elaborate.

Modified in background introduction and discussion analysis.The results show that biomass sorghum can tolerate relatively high concentration of Cd and has the ability to inhibit the transfer of Cd to the aboveground part of plants. In this discussion, we analyzed the possible molecular mechanisms, which can be used as reference for application and reference in other plants. In short, it is concluded that biomass sorghum can enrich Cd in the root part and has a good growth. In addition, it is not a food crop, so this study concluded that it is suitable for soil Cd repair plant.

Reviewer 2 Report

The manuscript under title "RNA-seq analysis of aboveground and underground parts of biomass sorghum was performed to evaluate its suitability for environmental remediation" is very important for the field and well written. Just I have some notice about the plant treatment section:

the authors mentioned that "After culturing for 14 days, well-growing biomass sorghum seedlings were selected and set 0, 15, 30, 60, 120 μmol·L-1 5 CdCl concentration medium stress for 7 days".

- the authors should clearly write where the treatments did? in  petri dishes or in pots? also the authors should write here how much they used from CdCl per-plant or per-pot

 - 2.3. Measurement of physiological indicators, the authors should write here at least a brief about each method for each indicator.

Author Response

the authors mentioned that "After culturing for 14 days, well-growing biomass sorghum seedlings were selected and set 0, 15, 30, 60, 120 μmol·L-1 5 CdCl concentration medium stress for 7 days".

- the authors should clearly write where the treatments did? in  petri dishes or in pots? also the authors should write here how much they used from CdCl per-plant or per-pot

After culturing for 14 days, well-growing biomass sorghum seedlings were selected and set 0, 15, 30, 60, 120 μmol·L-1 CdCl2  in  petri dishes hydroponic stress for 7 days.

 - 2.3. Measurement of physiological indicators, the authors should write here at least a brief about each method for each indicator.

Has been modified to:Determination of oxidase activity and MDA content